# The association of health insurance and race with treatment and survival in patients with metastatic colorectal cancer

Anastasios T. Mitsakos[1]⊕, William Irish[2,3]⊕, Alexander A. Parikh[1]⊕, Rebecca A. Snyder[1,3]⊕ *

1 Division of Surgical Oncology, Department of Surgery, Brody School of Medicine at East Carolina University, Greenville, North Carolina, United States of America, 2 Division of Surgical Research, Department of Surgery, Brody School of Medicine at East Carolina University, Greenville, North Carolina, United States of America, 3 Department of Public Health, Brody School of Medicine at East Carolina University, Greenville, North Carolina, United States of America

⊕ These authors contributed equally to this work.
* snyderre19@ecu.edu

**Data Availability Statement:** The data used in our study was obtained from the National Cancer Database (NCDB), specifically the Participant User File (PUF) 2016, containing patient cases with

## Abstract

### Background

Black patients and underinsured patients with colorectal cancer (CRC) present with more advanced disease and experience worse outcomes. The study aim was to evaluate the interaction of health insurance status and race with treatment and survival in metastatic CRC.

### Materials and methods

Patients diagnosed with metastatic CRC within NCDB from 2006–2016 were included. Primary outcomes included receipt of chemotherapy and 3-year all-cause mortality. Multivariable logistic regression and Cox-regression (MVR) including a two-way interaction term of race and insurance were performed to evaluate the differential association of race and insurance with receipt of chemotherapy and mortality, respectively.

### Results

128,031 patients were identified; 70.6% White, 14.4% Black, 5.7% Hispanic, and 9.3% Other race. Chemotherapy use was higher among White compared to Black patients. 3-year mortality rate was higher for Blacks and lower for Hispanics, in comparison with White patients. By MVR, Black patients were less likely to receive chemotherapy. When stratified by insurance status, Black patients with private and Medicare insurance were less likely to receive chemotherapy than White patients. All-cause mortality was higher in Black patients and lower in Hispanic patients, and these differences persisted after controlling for insurance and receipt of chemotherapy.

colon and rectal cancer diagnosed in 2004-2016. The specific data available from the NCDB are used under license for the current study, and thus are not publicly available. The American College of Surgeons Participant User Data File is available at: https://ncdbapp.facs.org/puf/. The data used for this study are third-party (i.e., data not owned or collected by the authors). The data can be accessible by other institutions or physicians in the same way as the authors. The authors do not have any special privileges in order to acquire these data.

**Funding:** The authors received no specific funding for this work.

**Competing interests:** The authors have declared that no competing interests exist.

## Conclusion

Black patients and uninsured or under-insured patients with metastatic CRC are less likely to receive chemotherapy and have increased mortality. The effect of health insurance among Blacks and Whites differs, however, and improving insurance alone does not appear to fully mitigate racial disparities in treatment and outcomes.

## Introduction

Colorectal cancer (CRC) remains one of the most common cancers in the United States and worldwide, accounting for more than 53,000 deaths, or 8.8% of all cancer deaths in the United States in 2020 [1]. Prior literature has demonstrated that the incidence of CRC among Black patients is higher than any other ethnic or racial group [2–4]. In addition, Black patients with CRC suffer from worse overall and stage-specific survival compared with White patients [3,5–8].

Despite major advances in screening and early treatment options in CRC, Black patients present with more advanced disease compared with other racial groups [5,9,10]. Racial disparities in treatment delivery extend from early-stage to late-stage disease and have a significant effect on overall mortality for Black patients [11–13]. Several studies have demonstrated that Black patients with advanced locoregional and metastatic disease are less likely to receive chemotherapy and surgery when clinically indicated [10,14–18]. The effects of treatment differences on disparities in survival may be greater in those patients with advanced compared to early-stage CRC [19].

Additionally, several studies have indicated that health insurance may at least partially mitigate disparities in survival between Black and White patients with CRC [20–23]. Several large, population-based retrospective studies using the Surveillance, Epidemiology, and End Results (SEER) database and the National Cancer Database (NCDB) have demonstrated that lack of insurance is associated with worse survival in CRC [21,23]. However, the effect of health insurance status on racial disparities in treatment delivery and long-term outcomes in patients with metastatic colorectal cancer has not been examined.

The primary aim of this study was to evaluate the interaction of health insurance status and race with receipt of chemotherapy and overall mortality in patients with metastatic CRC. The hypothesis was that racial disparities in receipt of chemotherapy and mortality would persist even when health insurance status was equivalent.

## Materials and methods

### Data source

A retrospective cohort study was performed using the National Cancer Database (NCDB), a clinical oncology database jointly sponsored by the American College of Surgeons and the American Cancer Society and sourced from hospital registry data collected by more than 1,500 Commission on Cancer (CoC)-accredited facilities. Data represent more than 70% of newly diagnosed cancer cases nationwide and more than 34,000,000 historical records [24].

All patients who were diagnosed with colon and rectal cancer were identified from the NCDB Participant User File Dataset. Race was defined as White, Black, Hispanic, or Other (American Indian, Eskimo, Asian subcategories, Pacific Islander) according to predefined NCDB categories. Of these, only patients with a primary malignancy diagnosis of stage IV

CRC diagnosed between 2006 and 2016 were selected. Patients for whom receipt of chemotherapy was unknown or with inadequate follow-up data were excluded from the final cohort. The study protocol was reviewed by the East Carolina University / Vidant Medical Center Institutional Review Board and was determined to be exempt.

## Variables and outcomes

Patient demographic and clinical variables assessed included: sex, age, race, treatment facility type, distance of patient from facility, insurance status, median income, education level (based on high school degree completion), region of residence, pathological grade, primary site of CRC, and Charlson-Deyo comorbidity index. Health insurance status was identified as the patient's primary insurance carrier at the time of diagnosis and/or treatment. For patients with more than one payer or insurance carrier, only the first insurance type was recorded. Distance traveled for treatment was calculated based on the patient's residential ZIP Code and street address of the treatment facility. Median household income was estimated based on patient ZIP Code using the 2012 American Community Survey data (2008–2012) and adjusting for 2012 inflation. Median household income was categorized into quartiles based on equally proportioned and representative income ranges corresponding to US ZIP Codes. Educational attainment was similarly estimated based on the number of adults in the patient's ZIP Code who did not graduate from high school and categorized as equally proportioned quartiles among all US ZIP Codes. Data were collected regarding receipt of treatment, including systemic chemotherapy, palliative care, as well as overall mortality. All collected data were obtained from pre-defined NCDB variables [25].

## Statistical analysis

Continuous variables are summarized overall and by race by presenting the mean and standard deviation (SD) or median and inter-quartile range (IQR– 25th and 75th percentile), while categorical variables are summarized by presenting counts and percentages, overall and by race. Continuous variables were compared between groups using the standard two-sample t-test or Wilcoxon Rank Sum test, where applicable. Categorical variables were compared between groups using Chi-square test.

Multivariable binary logistic regression analysis was performed to evaluate the association of race with the probability of receiving systemic chemotherapy. Two models were fit to the data: 1) Main effects model with additive terms for race and insurance status, adjusted for additional covariates and 2) Joint effects model with two-way interaction term for race and insurance status, adjusted for additional covariates. The latter model was used to evaluate the effect of race on the probability of receiving systemic chemotherapy within levels of insurance status. The likelihood ratio chi-square statistic was used to test the significance of the two-way interaction term. Covariates included: age, race, sex, insurance status, treatment facility type, income level, education, rurality, comorbidity, distance traveled for care, and tumor grade. Adjusted odds ratio (OR) and 95% confidence interval (CI) are provided as measures of strength of association and precision, respectively.

Patient survival was estimated using the life-table method. Multivariable Cox hazard model was used to evaluate the association of race with risk of all-cause mortality, adjusting for age, race, sex, insurance status, treatment facility type, income level, education, rurality, comorbidity, distance traveled for care, and tumor grade. Both the main effects model and joint effects model as described above were also used for this analysis. The latter model was used to evaluate the effect of race on the hazard of death within levels of insurance status. The likelihood ratio chi-square statistic was used to test the significance of the two-way interaction term.

Proportional hazards assumption was evaluated using numerical and graphical techniques. Hazard ratio and 95% confidence interval are provided as measures of strength of association and precision, respectively. P-value <0.05 was considered statistically significant. Statistical analyses were performed using SAS statistical software (version 9.4, SAS Institute, Cary, NC).

## Results

### Demographics

A total of 1,002,621 patients with newly diagnosed colorectal cancer from 2006–2016 were identified within the NCDB dataset. Patients with a different primary malignant diagnosis, stage I-III disease, unknown receipt of chemotherapy, and missing follow-up data were excluded. (Fig 1) The final cohort included 128,031 patients with metastatic colorectal cancer, of whom 70.6% (n = 90,382) were non-Hispanic White, 14.4% (n = 18,407) non-Hispanic Black, 5.7% (n = 7,340) Hispanic, and 9.3% (n = 11,902) Other race.

White patients were more commonly insured through private insurance or Medicare compared to Black and Hispanic patients, while Hispanic patients were more often Medicaid-insured or uninsured. (Table 1) Black and Hispanic patients had lower annual income, lower level of education, and more often lived in a metropolitan rather than an urban area when compared to White patients. (Table 1) White patients were more often treated at a community cancer program or comprehensive community cancer program, while their Black and Hispanic counterparts received care more often at academic/research programs, which includes NCI-designated comprehensive cancer centers. (Table 1) Pathological grade and Charlson-Deyo comorbidity index were similar among patients of all races (Table 1).

### Receipt of treatment

On unadjusted analysis, rates of systemic chemotherapy were slightly higher among non-Hispanic White compared to non-Hispanic Black patients with metastatic CRC (69.5% vs. 67.5%). Rates of receipt of palliative care in non-Hispanic White patients were similar to Black patients (12.7% vs. 12.3%). Hispanic patients had a higher rate of receipt of systemic chemotherapy (71.0%), but lower rate of receipt of palliative care (10.6%) when compared to non-Hispanic White and non-Hispanic Black patients (Table 2).

On adjusted analysis, non-Hispanic Black patients had a significantly lower odds of receiving systemic chemotherapy compared to non-Hispanic White patients (OR 0.82; 95% CI 0.78–0.85). Hispanic patients had a similar odds of receipt of systemic chemotherapy compared to White patients (OR 0.94; 95% CI 0.89–1.00). Health insurance status other than private/managed care was also associated with decreased odds of receiving chemotherapy (OR 0.88; 95% CI 0.84–0.91 for Medicare, OR 0.62; 95% CI 0.59–0.66 for Medicaid, and OR 0.48; 95% CI 0.45–0.50 for uninsured patients). Other factors independently associated with lower odds of receipt of chemotherapy in patients with metastatic CRC were higher Charlson-Deyo comorbidity index, lower median annual income, and lower educational status (Table 3A).

The two-way interaction term for race X insurance status in the joint effects model was statistically significant (LR statistic = 54.20; p<0.0001 on 15 degrees of freedom); suggesting that the effect of race on receipt of chemotherapy is differentially affected by type of insurance. When stratified by insurance status, non-Hispanic Black patients with private insurance or Medicare had lower odds of receiving chemotherapy compared to non-Hispanic White patients within the same insurance categories (OR 0.72; 95% CI 0.67–0.78 and OR 0.81; 95% CI 0.77–0.86, respectively). In patients with Medicaid, other government insurance, or no insurance at all, no significant difference in the receipt of chemotherapy was observed between non-Hispanic Black and non-Hispanic White patients (Table 3B).

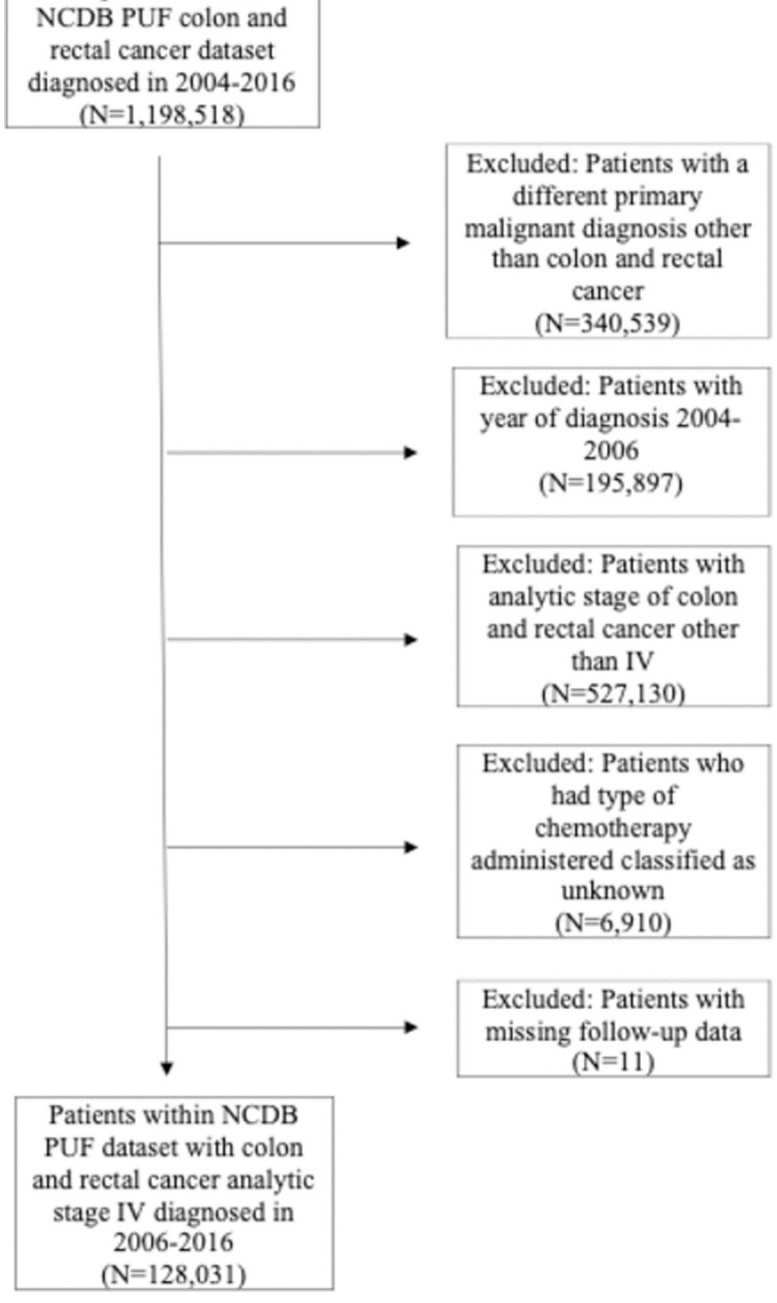

**Fig 1. Flowchart diagram of the selection of cohort for analysis.**

## All-cause mortality

Overall median follow-up was 62 months (IQR 37–92 months). Unadjusted patient survival at 3 years was 23.1% for non-Hispanic White patients, 20.3% for non-Hispanic Black patients, 30.5% for Hispanic patients, and 23.1% for patients of Other race. (Fig 2) On adjusted analysis, non-Hispanic Black patients had increased risk of overall mortality compared to non-Hispanic White patients, with and without controlling for receipt of systemic chemotherapy (adjusted HR 1.06; 95% CI 1.04–1.08 and HR 1.09; 95% CI 1.07–1.11, respectively), while Hispanic

**Table 1. Social, demographic, and clinical characteristics by race.**

| | Total Cohort N = 128,031 (100.0%) | Non-Hispanic White N = 90,382 (70.6%) | Non-Hispanic Black N = 18,407 (14.4%) | Hispanic N = 7,340 (5.7%) | Other N = 11,902 (9.3%) |
|---|---|---|---|---|---|
| **Age** | | | | | |
| Mean years (± SD) | 63.5 (± 14.1) | 64.4 (± 14.0) | 61.2 (± 13.5) | 59.6 (± 14.6) | 63.2 (± 14.3) |
| **Sex** | | | | | |
| Male | 65,876 (51.5%) | 46,760 (51.7%) | 8,916 (48.4%) | 4,039 (55.0%) | 6,161 (51.8%) |
| Female | 62,155 (48.5%) | 43,622 (48.3%) | 9,491 (51.6%) | 3,301 (45.0%) | 5,741 (48.2%) |
| **Treatment Facility** | | | | | |
| Community Cancer Program | 13,939 (10.9%) | 10,654 (11.8%) | 1,534 (8.3%) | 596 (8.1%) | 1,155 (9.7%) |
| Comprehensive Community Cancer Program | 50,961 (39.8%) | 38,112 (42.2%) | 5,888 (32.0%) | 2,376 (32.4%) | 4,585 (38.5%) |
| Academic/Research Program | 40,775 (31.9%) | 26,335 (29.1%) | 7,567 (41.1%) | 2,847 (38.8%) | 4,026 (33.8%) |
| Integrated Network Cancer Program | 16,752 (13.0%) | 11,827 (13.1%) | 2,526 (13.8%) | 870 (11.9%) | 1,529 (12.9%) |
| Unknown | 5,604 (4.4%) | 3,454 (3.8%) | 892 (4.8%) | 651 (8.9%) | 607 (5.1%) |
| **Insurance Status** | | | | | |
| Uninsured | 7,348 (5.7%) | 3,904 (4.3%) | 1,751 (9.5%) | 1,019 (13.9%) | 674 (5.7%) |
| Private Insurance / Managed Care | 49,475 (38.6%) | 36,099 (39.9%) | 6,158 (33.5%) | 2,450 (33.4%) | 4,768 (40.1%) |
| Medicaid | 11,243 (8.8%) | 5,928 (6.6%) | 2,820 (15.3%) | 1,341 (18.3%) | 1,154 (9.7%) |
| Medicare | 56,016 (43.8%) | 41,956 (46.4%) | 7,009 (38.1%) | 2,243 (30.6%) | 4,808 (40.4%) |
| Other Government | 1,339 (1.1%) | 887 (1.0%) | 222 (1.2%) | 59 (0.8%) | 171 (1.4%) |
| Insurance Status Unknown | 2610 (2.0%) | 1,608 (1.8%) | 447 (2.4%) | 228 (3.1%) | 327 (2.7%) |
| **Median Income Quartile** | | | | | |
| Less than $38,000 | 25,675 (20.1%) | 13,541 (15.0%) | 8,068 (43.8%) | 2,030 (27.7%) | 2,036 (17.1%) |
| $38,000 to $47,999 | 30,745 (24.0%) | 22,110 (24.5%) | 4,188 (22.7%) | 1,801 (24.5%) | 2,646 (22.2%) |
| $48,000 to $62,999 | 33,279 (26.0%) | 24,924 (27.6%) | 3,329 (18.1%) | 1,886 (25.7%) | 3,140 (26.4%) |
| $63,000 + | 37,785 (29.5%) | 29,414 (32.5%) | 2,756 (15.0%) | 1,588 (21.6%) | 4,027 (33.8%) |
| Not Available | 547 (0.4%) | 393 (0.4%) | 66 (0.4%) | 35 (0.5%) | 53 (0.5%) |
| **Percent Without High School Degree** | | | | | |
| 21% or more | 25,119 (19.6%) | 12,690 (14.0%) | 6,637 (36.1%) | 3,670 (50.0%) | 2,122 (17.8%) |
| 13% to 20.9% | 34,676 (27.1%) | 23,437 (25.9%) | 6,470 (35.1%) | 1,680 (22.9%) | 3,089 (26.0%) |
| 7% to 12.9% | 40,559 (31.7%) | 31,697 (35.1%) | 3,715 (20.2%) | 1,334 (18.2%) | 3,813 (32.0%) |
| Less than 7% | 27,198 (21.2%) | 22,215 (24.6%) | 1,526 (8.3%) | 624 (8.5%) | 2,833 (23.8%) |
| Not Available | 479 (0.4%) | 343 (0.4%) | 59 (0.3%) | 32 (0.4%) | 45 (0.4%) |
| **Distance Traveled for Care (miles)** | | | | | |
| Median (25[th]-75[th]) | 8.8 (3.9–21.1) | 9.8 (4.2–24.0) | 6.4 (3.0–13.7) | 6.9 (3.4–14.1) | 7.9 (3.8–18.6) |
| **Primary Site[a]** | | | | | |
| Right colon | 54,379 (42.5%) | 38,854 (43.0%) | 8,274 (45.0%) | 2,627 (35.8%) | 4,624 (38.9%) |
| Left colon | 34,802 (27.2%) | 23,865 (26.4%) | 5,069 (27.5%) | 2,245 (30.6%) | 3,623 (30.4%) |
| Other colon or Rectum | 38,850 (30.3%) | 27,663 (30.6%) | 5,064 (27.5%) | 2,468 (33.6%) | 3,655 (30.7%) |
| **Region** | | | | | |
| Metro | 104,639 (81.7%) | 71,600 (79.2%) | 16,331 (88.7%) | 6,810 (92.8%) | 9,898 (83.2%) |
| Urban | 17,836 (14.0%) | 14,424 (16.0%) | 1,564 (8.5%) | 356 (4.8%) | 1,492 (12.5%) |
| Rural | 2,475 (1.9%) | 1,952 (2.2%) | 225 (1.2%) | 30 (0.4%) | 268 (2.3%) |
| Not Available | 3,081 (2.4%) | 2,406 (2.7%) | 287 (1.6%) | 144 (2.0%) | 244 (2.0%) |
| **Charlson-Deyo Comorbidity Index** | | | | | |

*(Continued)*

**Table 1.** (Continued)

| | Total Cohort N = 128,031 (100.0%) | Non-Hispanic White N = 90,382 (70.6%) | Non-Hispanic Black N = 18,407 (14.4%) | Hispanic N = 7,340 (5.7%) | Other N = 11,902 (9.3%) |
|---|---|---|---|---|---|
| 0 | 95,505 (74.6%) | 67,468 (74.6%) | 13,369 (72.6%) | 5,645 (76.9%) | 9,023 (75.8%) |
| 1 | 23,305 (18.2%) | 16,364 (18.1%) | 3,617 (19.7%) | 1,260 (17.2%) | 2,064 (17.3%) |
| 2 | 6,161 (4.8%) | 4,425 (4.9%) | 928 (5.0%) | 267 (3.6%) | 541 (4.6%) |
| 3 or more | 3,060 (2.4%) | 2,125 (2.4%) | 493 (2.7%) | 168 (2.3%) | 274 (2.3%) |
| **Tumor Grade** | | | | | |
| 1 | 6,910 (5.4%) | 4,753 (5.3%) | 1,056 (5.7%) | 457 (6.2%) | 644 (5.4%) |
| 2 | 56,700 (44.3%) | 39,495 (43.7%) | 8,512 (46.2%) | 3,354 (45.7%) | 5,339 (44.9%) |
| 3 | 26,403 (20.6%) | 19,311 (21.4%) | 3,069 (16.7%) | 1,423 (19.4%) | 2,600 (21.8%) |
| 4 | 3,319 (2.6%) | 2,617 (2.9%) | 295 (1.6%) | 156 (2.1%) | 251 (2.1%) |
| Not Available | 34,699 (27.1%) | 24,206 (26.8%) | 5,475 (29.7%) | 1,950 (26.6%) | 3,068 (25.8%) |

SD: Standard deviation.

[a]Right colon: Includes cecum, appendix, ascending colon, hepatic flexure of colon, transverse colon; Left colon: Includes splenic flexure of colon, descending colon, sigmoid colon; Other colon or Rectum: Includes other overlapping lesion in the colon, rectum, or not otherwise specified.

patients had decreased risk of mortality (adjusted HR 0.81; 95% CI 0.78–0.83 and HR 0.83; 95% CI 0.81–0.85, respectively). (Table 4A) Health insurance status other than private/managed care was also associated with increased 3-year overall mortality independent of receipt of systemic chemotherapy. Risk of death was highest among uninsured patients with and without adjustment for chemotherapy (HR 1.27; 95% CI 1.23–1.30 and HR 1.32; 95% CI 1.29–1.36, respectively). (Table 4A) The two-way interaction term for race by insurance status was statistically significant for the joint effects models with or without adjusting for receipt of systematic chemotherapy (With adjustment: LR statistic = 39.97, p = 0.0005; without adjustment: LR statistic = 44.41, p<0.0001). The effect of race on all-cause mortality was more pronounced in patients with Private insurance or on Medicare. (Table 4B)

## Discussion

In this study of a large, national population of patients with metastatic CRC, Black patients, as well as those with Medicaid or no insurance, had lower rates for receipt of chemotherapy and higher 3-year overall mortality compared with White patients and those with private insurance. These findings are consistent with a number of prior studies showing significant racial

**Table 2. Receipt of chemotherapy and receipt of palliative care by race.**

| | Total Cohort N = 128,031 (100.0%) | Non-Hispanic White N = 90,382 (70.6%) | Non-Hispanic Black N = 18,407 (14.4%) | Hispanic N = 7,340 (5.7%) | Other N = 11,902 (9.3%) |
|---|---|---|---|---|---|
| **Receipt of Chemotherapy** | | | | | |
| No | 39,391 (30.8%) | 27,605 (30.5%) | 5,984 (32.5%) | 2,128 (29.0%) | 3,674 (30.9%) |
| Yes | 88,640 (69.2%) | 62,777 (69.5%) | 12,423 (67.5%) | 5,212 (71.0%) | 8,228 (69.1%) |
| **Receipt of Palliative Care** | | | | | |
| No | 111,553 (87.1%) | 78,472 (86.8%) | 16,068 (87.3%) | 6,505 (88.6%) | 10,508 (88.2%) |
| Yes | 15,887 (12.4%) | 11,501 (12.7%) | 2,260 (12.3%) | 774 (10.6%) | 1,352 (11.4%) |
| Unknown | 591 (0.5%) | 409 (0.5%) | 79 (0.4%) | 61 (0.8%) | 42 (0.4%) |

**Table 3A. Adjusted odds ratio of receipt of systemic chemotherapy using the main effects additive model.**

| | Receipt of Chemotherapy | |
|---|---|---|
| | **Odds Ratio** | **95% CI** |
| **Race (ref = Non-Hispanic White)** | | |
| Non-Hispanic Black | 0.82 | 0.78–0.85 |
| Hispanic | 0.94 | 0.89–1.00 |
| Other | 0.92 | 0.88–0.96 |
| **Insurance Status (ref = Private)** | | |
| Uninsured | 0.48 | 0.45–0.50 |
| Medicare | 0.88 | 0.84–0.91 |
| Medicaid | 0.62 | 0.59–0.66 |
| Other Government | 0.75 | 0.65–0.85 |
| Insurance Status Unknown | 0.68 | 0.62–0.75 |
| **Age at Diagnosis** | | |
| Per year increase | 0.932 | 0.931–0.934 |
| **Sex (ref = Male)** | | |
| Female | 0.92 | 0.89–0.94 |
| **Treatment Facility (ref = Comprehensive Community Cancer Program)** | | |
| Community Cancer Program | 0.94 | 0.90–0.98 |
| Academic/Research Program | 1.27 | 1.23–1.31 |
| Integrated Network Cancer Program | 0.99 | 0.95–1.03 |
| Unknown | 0.39 | 0.36–0.43 |
| **Charlson-Deyo Comorbidity Index (ref = 0)** | | |
| 1 | 0.82 | 0.80–0.85 |
| 2 | 0.61 | 0.57–0.64 |
| 3 or more | 0.41 | 0.38–0.45 |
| **Median Income Quartile (ref = Less than $38,000)** | | |
| $38,000 to $47,999 | 1.09 | 1.04–1.13 |
| $48,000 to $62,999 | 1.11 | 1.06–1.16 |
| $63,000 + | 1.14 | 1.08–1.20 |
| Not available | 1.37 | 0.75–2.48 |
| **Percent Without High School Degree (ref = 21% or more)** | | |
| 13% to 20.9% | 1.12 | 1.07–1.16 |
| 7% to 12.9% | 1.17 | 1.12–1.22 |
| Less than 7% | 1.24 | 1.18–1.31 |
| Not Available | 0.74 | 0.40–1.40 |
| **Distance Traveled for Care** | | |
| Per 50-mile increase | 0.994 | 0.988–0.999 |
| **Region (ref = Metro)** | | |
| Urban | 1.18 | 1.13–1.23 |
| Rural | 1.19 | 1.08–1.31 |
| Not Available | 1.01 | 0.93–1.11 |
| **Tumor Grade (ref = 1)** | | |
| 2 | 1.52 | 1.43–1.61 |
| 3 | 1.17 | 1.10–1.24 |
| 4 | 1.11 | 1.00–1.22 |
| Not Available | 0.76 | 0.72–0.81 |

CI: Confidence interval.

**Table 3B. Adjusted odds ratio of receipt of systemic chemotherapy among non-hispanic black, hispanic, and other race compared to non-hispanic white patients stratified on insurance status based on the joint effects model[a].**

| Insurance Status | Race (ref = Non-Hispanic White) | Receipt of Chemotherapy |
|---|---|---|
| | | Odds Ratio (95% CI) |
| Private Insurance / Managed Care | Non-Hispanic Black | 0.72 (0.67–0.78) |
| | Hispanic | 0.79 (0.71–0.89) |
| | Other | 0.84 (0.78–0.92) |
| Medicaid | Non-Hispanic Black | 0.90 (0.81–1.01) |
| | Hispanic | 1.04 (0.89–1.20) |
| | Other | 1.01 (0.86–1.18) |
| Medicare | Non-Hispanic Black | 0.81 (0.77–0.86) |
| | Hispanic | 0.98 (0.90–1.08) |
| | Other | 0.94 (0.88–1.00) |
| Other Government | Non-Hispanic Black | 0.89 (0.62–1.28) |
| | Hispanic | 1.01 (0.52–1.97) |
| | Other | 1.17 (0.77–1.79) |
| Uninsured | Non-Hispanic Black | 1.12 (0.98–1.28) |
| | Hispanic | 1.17 (0.99–1.38) |
| | Other | 1.05 (0.87–1.26) |
| Insurance Status Unknown | Non-Hispanic Black | 0.70 (0.55–0.90) |
| | Hispanic | 0.81 (0.57–1.14) |
| | Other | 0.79 (0.59–1.05) |

CI: Confidence interval.

[a]Two-way interaction term: Race X Insurance status, Likelihood Ratio statistic = 54.20, $p<0.0001$ on 15 degrees of freedom.

disparities in the receipt of multi-modality therapy and long-term outcomes [3,5–8,10,14–18]. While several studies have also delineated the positive impact of integrated health care systems and the importance of health insurance in early identification and receipt of appropriate treatment in CRC [20–23], the effect of health insurance status on racial disparities in receipt of therapy in metastatic CRC is not well understood. To our knowledge, this is the first study to specifically evaluate the interaction between race and health insurance in treatment and

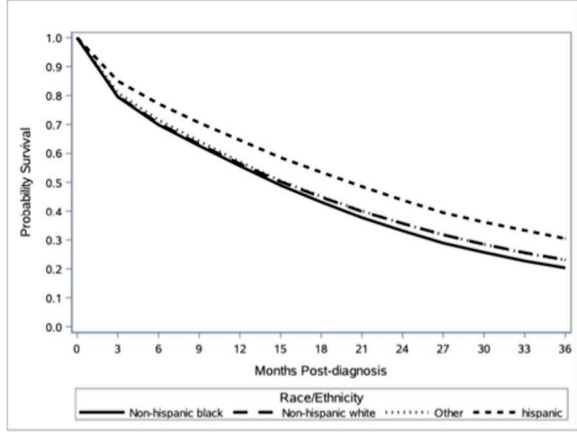

**Fig 2. 3-Year overall unadjusted patient survival curves by race using the life-table method.**

**Table 4A. Multivariable Cox hazards main effects model for all-cause mortality with and without adjustment for receipt of chemotherapy.**

| | Overall mortality without adjustment for chemotherapy | Overall mortality with adjustment for chemotherapy |
|---|---|---|
| | Hazard Ratio (95% CI) | Hazard Ratio (95% CI) |
| **Race (ref = Non-Hispanic White)** | | |
| Non-Hispanic Black | 1.09 (1.07–1.11) | 1.06 (1.04–1.08) |
| Hispanic | 0.83 (0.81–0.85) | 0.81 (0.78–0.83) |
| Other | 1.01 (0.98–1.03) | 0.99 (0.97–1.01) |
| **Insurance (ref = Private)** | | |
| Uninsured | 1.32 (1.29–1.36) | 1.27 (1.23–1.30) |
| Medicare | 1.08 (1.06–1.10) | 1.11 (1.09–1.13) |
| Medicaid | 1.28 (1.25–1.31) | 1.24 (1.21–1.27) |
| Other Government | 1.06 (0.99–1.12) | 1.03 (0.97–1.09) |
| Insurance Status Unknown | 1.25 (1.20–1.31) | 1.20 (1.14–1.25) |
| **Age at Diagnosis** | | |
| Per year increase | 1.027 (1.027–1.028) | 1.018 (1.017–1.019) |
| **Sex (ref = Male)** | | |
| Female | 0.97 (0.96–0.98) | 0.95 (0.94–0.96) |
| **Treatment Facility (ref = Comprehensive Community Cancer Program)** | | |
| Community Cancer Program | 1.03 (1.01–1.05) | 1.03 (1.01–1.05) |
| Academic/Research Program | 0.84 (0.83–0.85) | 0.86 (0.85–0.87) |
| Integrated Network Cancer Program | 0.98 (0.96–1.00) | 0.98 (0.96–1.00) |
| Unknown | 1.46 (1.40–1.51) | 1.24 (1.19–1.28) |
| **Charlson-Deyo Comorbidity Index (ref = 0)** | | |
| 1 | 1.10 (1.09–1.12) | 1.09 (1.07–1.11) |
| 2 | 1.29 (1.26–1.33) | 1.21 (1.18–1.25) |
| 3 or more | 1.69 (1.63–1.75) | 1.53 (1.47–1.59) |
| **Median Income Quartile (ref = Less than $38,000)** | | |
| $38,000 to $47,999 | 0.96 (0.94–0.98) | 0.97 (0.95–0.99) |
| $48,000 to $62,999 | 0.93 (0.91–0.95) | 0.93 (0.91–0.95) |
| $63,000 + | 0.89 (0.87–0.91) | 0.88 (0.86–0.91) |
| Not Available | 0.82 (0.63–1.08) | 0.88 (0.67–1.15) |
| **Percent Without High School Degree (ref = 21% or more)** | | |
| 13% to 20.9% | 1.01 (0.99–1.03) | 1.02 (1.01–1.04) |
| 7% to 12.9% | 1.00 (0.98–1.02) | 1.04 (1.01–1.06) |
| Less than 7% | 0.99 (0.97–1.02) | 1.03 (1.00–1.05) |
| Not Available | 1.07 (0.80–1.43) | 0.99 (0.74–1.32) |
| **Distance Traveled for Care** | | |
| Per 50-mile increase | 0.985 (0.982–0.988) | 0.984 (0.981–0.987) |
| **Region (ref = Metro)** | | |
| Urban | 0.99 (0.97–1.01) | 1.01 (0.99–1.03) |
| Rural | 1.01 (0.96–1.05) | 1.04 (0.99–1.08) |
| Not Available | 1.00 (0.96–1.04) | 0.99 (0.95–1.03) |
| **Tumor Grade (ref = 1)** | | |
| 2 | 1.25 (1.21–1.29) | 1.38 (1.34–1.42) |
| 3 | 1.91 (1.85–1.97) | 2.11 (2.05–2.18) |
| 4 | 1.98 (1.89–2.08) | 2.18 (2.08–2.28) |
| Not Available | 2.20 (2.13–2.27) | 2.31 (2.23–2.38) |
| **Chemotherapy (ref = No)** | | |

*(Continued)*

**Table 4A.** (Continued)

| | Overall mortality without adjustment for chemotherapy | Overall mortality with adjustment for chemotherapy |
|---|---|---|
| | Hazard Ratio (95% CI) | Hazard Ratio (95% CI) |
| Yes | n/a | 0.443 (0.437–0.449) |

CI: Confidence interval; n/a = not applicable

**Table 4B. Adjusted hazards ratio of all-cause mortality among black, hispanic, and other race compared to white patients stratified on insurance status based on the joint effects model[a].**

| Insurance Status | Race (ref = Non-Hispanic White) | Overall mortality without adjustment for chemotherapy | Overall mortality with adjustment for chemotherapy |
|---|---|---|---|
| | | Hazard Ratio (95% CI) | Hazard Ratio (95% CI) |
| Private Insurance/Managed Care | Non-Hispanic Black | 1.10 (1.07–1.14) | 1.08 (1.04–1.11) |
| | Hispanic | 0.88 (0.84–0.93) | 0.86 (0.82–0.90) |
| | Other | 1.05 (1.01–1.08) | 1.03 (0.99–1.06) |
| Medicaid | Non-Hispanic Black | 1.04 (0.99–1.09) | 1.02 (0.97–1.07) |
| | Hispanic | 0.78 (0.73–0.84) | 0.75 (0.70–0.81) |
| | Other | 0.92 (0.86–0.99) | 0.91 (0.85–0.98) |
| Medicare | Non-Hispanic Black | 1.10 (1.07–1.13) | 1.07 (1.04–1.10) |
| | Hispanic | 0.81 (0.77–0.85) | 0.79 (0.75–0.83) |
| | Other | 1.00 (0.97–1.03) | 0.99 (0.96–1.02) |
| Other Government | Non-Hispanic Black | 1.04 (0.88–1.23) | 1.03 (0.87–1.22) |
| | Hispanic | 1.13 (0.83–1.53) | 1.15 (0.85–1.55) |
| | Other | 1.05 (0.87–1.26) | 1.05 (0.88–1.27) |
| Uninsured | Non-Hispanic Black | 1.00 (0.94–1.06) | 1.01 (0.95–1.08) |
| | Hispanic | 0.74 (0.68–0.80) | 0.73 (0.67–0.80) |
| | Other | 0.91 (0.83–1.00) | 0.91 (0.83–1.00) |
| Insurance Status Unknown | Non-Hispanic Black | 1.10 (0.98–1.23) | 1.01 (0.90–1.14) |
| | Hispanic | 0.97 (0.83–1.14) | 0.95 (0.81–1.11) |
| | Other | 0.95 (0.83–1.09) | 0.90 (0.79–1.03) |

CI: Confidence interval.

[a]The two-way interaction term for Race X Insurance status was only statistically significant in the multivariable Cox hazard model without adjustment for chemotherapy, Likelihood ratio chi square = 45.41, p<0.0001 on 15 degrees of freedom.

outcomes of patients with metastatic CRC. This study demonstrated that even among patients with private or Medicare insurance, Black patients with metastatic CRC are less likely to receive chemotherapy. Further, although Black patients have increased overall mortality independent of health insurance status and receipt of chemotherapy, the lack of adequate health insurance, independent of race, had a greater impact on both receipt of chemotherapy and mortality.

Despite advances in both systemic chemotherapy and surgical management of metastatic CRC [26–28], disparities persist across racial and demographic groups. Prior SEER studies have demonstrated that although cancer-specific and overall survival have improved over time, White patients with metastatic CRC have experienced more marked improvements compared to Black patients, suggesting that treatment delivery may differ by race [14,29]. In one study of the linked SEER-Medicare database, rates of specialist consultation and subsequent

treatment with multimodality therapy were lower for Black patients with metastatic CRC [15]. More recently, a California Cancer Registry study of patients with colorectal liver metastases demonstrated that Black patients had worse survival, lower rates of chemotherapy, and lower rates of liver resection when compared to their White counterparts [18]. Accordingly, the present study also demonstrated that Black patients with metastatic CRC are less likely to receive chemotherapy even within the "better insured" cohorts–i.e. private or Medicare. Although rates of chemotherapy were also lower within other insurance groups, the findings did not reach statistical significance. The number of patients within these cohorts was smaller, however, and therefore this analysis may not be adequately powered to observe a difference in these other insurance groups.

Although the association between health insurance and racial disparities in screening, stage of presentation, and survival in CRC has been previously explored in the literature, the role that these factors play in treatment delivery and survival specifically in stage IV CRC has not been well-described. In one study of a racial/ethnic minority population sample residing in low-income housing sites, no difference in CRC screening was observed between White and Black patients with the same insurance coverage [30]. Following the recent implementation of the Affordable Care Act, an NCDB study demonstrated that enrollment rates in primary therapies for stage IV CRC were more favorable for Black than White patients. Although only a single study, these findings suggest that policy changes may be efficacious in reducing racial disparities [22]. Findings from the present study suggest, however, that providing Black patients with better insurance alone may not be enough to adequately reduce disparities in the treatment of metastatic CRC.

Finally, this study demonstrated that patients who live in lower-income or less educated areas are also less likely to receive systemic chemotherapy and have increased overall mortality rates independent of race. Several studies have investigated the impact of social determinants of health on CRC care delivery. In one study, racial and economic segregation, defined as the extent to which an area's population is concentrated into extremes of deprivation and privilege was strongly associated with limited access to affordable health care and increased odds of advanced disease at diagnosis [9]. Results from a California Cancer Registry cohort study identified a survival benefit in patients residing in neighborhoods of higher socioeconomic status [31]. In contrast, in two studies of patients treated in safety-net hospitals and integrated health care systems with equal access to care, racial disparities were not observed [20,32]. According to two recent systematic reviews, focused interventions to address social determinants of health are needed to improve cost-effective colorectal cancer screening in underserved, vulnerable populations, since factors such as poverty, lack of education, immigration status, lack of social support, and social isolation play a significant role in stage at diagnosis and overall survival [33,34]. Even though race and insurance appear to play a significant role in CRC care delivery and mortality as illustrated in this study, these factors do not fully explain the existing disparities. Further investigation will be critical to better understand the complex interactions between social determinants of health and CRC treatment and outcomes and to design targeted interventions to address disparities.

This study has several limitations. First, this is a retrospective cohort study and is limited by the quality of data abstraction by NCDB registrars. Second, it is comprised of data from Commission-on-Cancer-accredited facilities and therefore may not be generalizable to other patient populations. Third, only patients with initial presentation of metastatic CRC could be assessed; therefore, these results may not be representative of treatment and survival rates in patients who develop metachronous metastatic disease. Fourth, information related to social determinants of health remains limited within the NCDB and other national cancer registries; therefore, the influence of additional socioeconomic and neighborhood factors on receipt of

treatment and outcomes remains poorly understood. Fifth, the racial composition of patients within the NCDB dataset differs from the general US population, which may limit generalizability. More specifically, the Hispanic cohort within this NCDB sample was 5.7%, which is lower than the 18.5% ratio of the Hispanic population in the general United States per the latest United States Census Bureau data [35]. Finally, NCDB lacks details regarding specific systemic chemotherapy regimens and/or biologic therapy agents and number of treatments; therefore, adherence to national guidelines and/or treatment compliance is unknown.

## Conclusions

In this observational cohort study of patients with metastatic CRC diagnosed between 2006 and 2016, Black patients were less likely to receive chemotherapy even when privately and Medicare insured. Racial disparities in receipt of chemotherapy were no longer observed in the subgroup of patients who were uninsured or who had Medicaid insurance, likely due to universal poor access to health care and other confounding social determinants of health prevalent within this patient population. Three-year overall mortality remained higher among Black patients, even after controlling for differences in health insurance status and receipt of chemotherapy. Although insured patients are more likely to receive appropriate treatment and experience better outcomes, health insurance does not appear to fully mitigate racial differences in survival and receipt of treatment. Therefore, simply providing better insurance to disadvantaged populations may not be enough to decrease these disparities. Other important factors, including social determinants of health, some of which were included within this study, such as literacy, socioeconomic status, education, and access to care likely contribute to these disparities and warrant further investigation to reduce ongoing racial disparities in health care.

## Author Contributions

**Conceptualization:** Anastasios T. Mitsakos, William Irish, Alexander A. Parikh, Rebecca A. Snyder.

**Data curation:** Anastasios T. Mitsakos, William Irish, Alexander A. Parikh, Rebecca A. Snyder.

**Formal analysis:** Anastasios T. Mitsakos, William Irish, Alexander A. Parikh, Rebecca A. Snyder.

**Investigation:** Anastasios T. Mitsakos, William Irish, Alexander A. Parikh, Rebecca A. Snyder.

**Methodology:** Anastasios T. Mitsakos, William Irish, Alexander A. Parikh, Rebecca A. Snyder.

**Project administration:** Anastasios T. Mitsakos, William Irish, Alexander A. Parikh, Rebecca A. Snyder.

**Resources:** Anastasios T. Mitsakos, William Irish, Alexander A. Parikh, Rebecca A. Snyder.

**Software:** Anastasios T. Mitsakos, William Irish, Alexander A. Parikh, Rebecca A. Snyder.

**Supervision:** Anastasios T. Mitsakos, William Irish, Alexander A. Parikh, Rebecca A. Snyder.

**Validation:** Anastasios T. Mitsakos, William Irish, Alexander A. Parikh, Rebecca A. Snyder.

**Visualization:** Anastasios T. Mitsakos, William Irish, Alexander A. Parikh, Rebecca A. Snyder.

**Writing – original draft:** Anastasios T. Mitsakos, William Irish, Alexander A. Parikh, Rebecca A. Snyder.

**Writing – review & editing:** Anastasios T. Mitsakos, William Irish, Alexander A. Parikh, Rebecca A. Snyder.

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
