## [Decision Letter · Decision Letter 0]

6 Aug 2021

PONE-D-21-14137

The Association of Health Insurance and Race with Treatment and Survival in Patients with Metastatic Colorectal Cancer

PLOS ONE

Dear Dr. Snyder,

Thank you for submitting your manuscript to PLOS ONE. After careful consideration, we feel that it has merit but does not fully meet PLOS ONE’s publication criteria as it currently stands. Therefore, we invite you to submit a revised version of the manuscript that addresses the points raised during the review process.

This manuscript addresses a very timely topic of race-related cancer disparities. At present, better understanding of the causal factors is needed to improve this public health care issue. This type of manuscript is crucial in furthering the general understanding of insurance related cancer disparities.

We look forward to receiving your revised manuscript.

Kind regards,

Curtis J. Wray, M.D., M.S.

Academic Editor

PLOS ONE

Journal Requirements:

Additional Editor Comments:

Thank you for this submission to PLOS ONE. This is an interesting manuscript; however, revisions are needed prior to acceptance and publication. The comments from the reviewers are attached. Please respond to the comments.

Reviewers' comments:

Reviewer's Responses to Questions

**Comments to the Author**

1. Is the manuscript technically sound, and do the data support the conclusions?

Reviewer #1: Yes

Reviewer #2: Yes

2. Has the statistical analysis been performed appropriately and rigorously? 

Reviewer #1: Yes

Reviewer #2: Yes

3. Have the authors made all data underlying the findings in their manuscript fully available?

Reviewer #1: Yes

Reviewer #2: Yes

4. Is the manuscript presented in an intelligible fashion and written in standard English?

Reviewer #1: Yes

Reviewer #2: Yes

5. Review Comments to the Author

Reviewer #1: Overall, this is a well written manuscript evaluating receipt of chemotherapy for mCRC influenced by race and insurance status using NCDB

1. Why were other races excluded from analysis? Please provide justification for this.

Reviewer #2: I appreciate the opportunity to review this paper. Mitsakos and colleagues presents a retrospective study of patients from the NCDB diagnosed with metastatic colorectal cancer from 2006-2016, and aimed to evaluate the interaction of health insurance status and race with treatment and survival. The primary outcome was receipt of chemotherapy and 3-year all-cause mortality. The found that Black patients were less likely to receive chemotherapy, had a higher 3-year mortality rate, and were less likely to receive chemotherapy even when stratified by insurance status as compared to White patients. Using their two-way interaction term analysis, they were able to show that insurance status alone did not account for the disparities in treatment and outcomes according to race. The finding that there are disparities in treatment for colorectal cancer by race is not novel, but what is novel is the finding that these disparities cannot be explained solely by insurance status or quality of insurance.

I have the following questions aimed at strengthening the manuscript.

-was “Receipt of chemotherapy” in the interaction analyses inclusive of palliative chemotherapy?

-Why did the authors choose to exclude the Hispanic or Other groups of patients for this paper? It would be interesting to know if similar findings applied to these groups as well, and what the impact of insurance status on treatments and outcomes are for them.

-what do the authors believe is the explanation for their findings that Black patients with “better” insurance (private, Medicare) are less likely to receive chemotherapy compared to White patients? The authors offer a hypothesis for their findings that under-insured groups have likely universally less access to care, thus explaining why this was not found in those insurance groups. Although, they state that the cohorts were smaller in the underinsured groups, and thus underpowered to observe a similar association with the “worse” insurance groups. What is their explanation for this finding?

-suggest adding to the discussion that although yes, the authors found that there are differences in receipt of chemotherapy based upon race, but the differences are actually small, perhaps smaller than they may have hypothesized. Because of such high numbers of patients in the NCDB, every comparison is essentially statistically significant if different by 0.1%.

-is the 3-year all cause mortality presented in Figure 2 statistically significant?

-In the methods the authors state they collected data including surgery, and combination chemo+surgery, but I do not see any mention of surgery in the tables, etc.

6. PLOS authors have the option to publish the peer review history of their article (what does this mean?). If published, this will include your full peer review and any attached files.

Reviewer #1: No

Reviewer #2: No

---

## [Author Response · Author response to Decision Letter 0]

15 Sep 2021

Reviewer 1

Overall, this is a well written manuscript evaluating receipt of chemotherapy for mCRC influenced by race and insurance status using NCDB.

1. Why were other races excluded from analysis? Please provide justification for this. 

Thank you very much for your comments and for this excellent question. Initially, our intention was to compare Non-Hispanic White and Non-Hispanic Black patients only, since most previous studies have noted the most concerning disparities between these two population groups. However, per your suggestion, we have now included all races (Non-Hispanic White, Non-Hispanic Black, Hispanic, Other) in our revised analysis. In our revised manuscript, all the tables, figures, and results, have been updated accordingly with the addition of the Hispanic and Other races in the cohort. 

Reviewer 2

I appreciate the opportunity to review this paper. Mitsakos and colleagues presents a retrospective study of patients from the NCDB diagnosed with metastatic colorectal cancer from 2006-2016, and aimed to evaluate the interaction of health insurance status and race with treatment and survival. The primary outcome was receipt of chemotherapy and 3-year all-cause mortality. The found that Black patients were less likely to receive chemotherapy, had a higher 3-year mortality rate, and were less likely to receive chemotherapy even when stratified by insurance status as compared to White patients. Using their two-way interaction term analysis, they were able to show that insurance status alone did not account for the disparities in treatment and outcomes according to race. The finding that there are disparities in treatment for colorectal cancer by race is not novel, but what is novel is the finding that these disparities cannot be explained solely by insurance status or quality of insurance.

I have the following questions aimed at strengthening the manuscript.

1. Was “Receipt of chemotherapy” in the interaction analyses inclusive of palliative chemotherapy?

Thank you for your comments and for this excellent question. According to the NCDB Participant User Data File (NCDB PUF), “Receipt of chemotherapy” includes any type of systemic chemotherapy administered to the patient at any point in time, with the exception of chemotherapy provided to control symptoms, alleviate pain, or make the patient more comfortable, which is then included in the item “Palliative Care”. Based on this definition, “Receipt of chemotherapy” in the interaction analyses is not inclusive of palliative chemotherapy. 

2. Why did the authors choose to exclude the Hispanic or Other groups of patients for this paper? It would be interesting to know if similar findings applied to these groups as well, and what the impact of insurance status on treatments and outcomes are for them.

Thank you for this thoughtful comment. As we also responded to Reviewer 1 (above), our intention was to compare Non-Hispanic White and Non-Hispanic Black patients only, since most previous studies have noted the most concerning disparities between these two population groups. However, per your suggestion, we have now included all races (Non-Hispanic White, Non-Hispanic Black, Hispanic, Other) in our revised analysis. All the tables, figures, and results have been updated accordingly with the addition of the Hispanic and Other races in the cohort. The Hispanic subcohort appears to be numerically under-represented in this NCDB dataset in comparison to US census data, which could limit generalizability. We have discussed this potential limitation in the Discussion. 

3. What do the authors believe is the explanation for their findings that Black patients with “better” insurance (private, Medicare) are less likely to receive chemotherapy compared to White patients? The authors offer a hypothesis for their findings that under-insured groups have likely universally less access to care, thus explaining why this was not found in those insurance groups. Although, they state that the cohorts were smaller in the underinsured groups, and thus underpowered to observe a similar association with the “worse” insurance groups. What is their explanation for this finding?

Thank you for this excellent comment which illustrates one of the major points of the study. Race, insurance status, and social determinants of health likely all play a complex role in colorectal cancer treatment delivery and mortality, and the relative contribution of each is difficult to characterize. Even when trying to isolate these factors by statistical analysis, it appears that just the improvement of health insurance status in racially disadvantaged populations cannot completely mitigate racial disparities in treatment and survival. We believe this illustrates that other underlying social determinants of health, such as general healthcare access, neighborhood segregation, or health literacy, for example, likely contribute to these disparities and warrant further exploration.

4. Suggest adding to the discussion that although yes, the authors found that there are differences in receipt of chemotherapy based upon race, but the differences are actually small, perhaps smaller than they may have hypothesized. Because of such high numbers of patients in the NCDB, every comparison is essentially statistically significant if different by 0.1%.

Thank you for this suggestion. It is an inherent limitation of our study, as with any retrospective study dealing with very large cohort samples, that univariate analyses can yield statistically significant differences due to a large sample size, but that these differences are not always clinically significant. For this reason, we elected to delete all p-values from Table 2 and from the univariate analysis of receipt of chemotherapy by race discussed in the Results. We hope that this allows the reader to focus on the adjusted multivariable analysis, which is more informative as it accounts for important confounding factors.

5. Is the 3-year all cause mortality presented in Figure 2 statistically significant?

Differences in 3-year all-cause mortality by race are statistically significant. However, as stated above, we did not report a log-rank analysis due to the large sample size and likelihood of a minor statistical significance in the absence of clinical significance. For this reason, we elected to present these findings as a Life-Table curve with the associated unadjusted all-cause mortality rates for each race. 

6. In the methods the authors state they collected data including surgery, and combination chemo+surgery, but I do not see any mention of surgery in the tables, etc. 

Thank you for recognizing and pointing out this error. In this study, we did not collect data regarding surgical therapy, but only systemic chemotherapy, palliative care, and mortality. We have corrected this in the Methods section.

---

## [Decision Letter · Decision Letter 1]

28 Jan 2022

The Association of Health Insurance and Race with Treatment and Survival in Patients with Metastatic Colorectal Cancer

PONE-D-21-14137R1

Dear Dr. Snyder,

We’re pleased to inform you that your manuscript has been judged scientifically suitable for publication and will be formally accepted for publication once it meets all outstanding technical requirements.

Kind regards,

Girijesh Kumar Patel, PhD

Academic Editor

PLOS ONE

Additional Editor Comments (optional):

Reviewers' comments:

Reviewer's Responses to Questions

**Comments to the Author**

1. If the authors have adequately addressed your comments raised in a previous round of review and you feel that this manuscript is now acceptable for publication, you may indicate that here to bypass the “Comments to the Author” section, enter your conflict of interest statement in the “Confidential to Editor” section, and submit your "Accept" recommendation.

Reviewer #1: All comments have been addressed

Reviewer #2: All comments have been addressed

2. Is the manuscript technically sound, and do the data support the conclusions?

Reviewer #1: Yes

Reviewer #2: Yes

3. Has the statistical analysis been performed appropriately and rigorously? 

Reviewer #1: Yes

Reviewer #2: Yes

4. Have the authors made all data underlying the findings in their manuscript fully available?

Reviewer #1: Yes

Reviewer #2: Yes

5. Is the manuscript presented in an intelligible fashion and written in standard English?

Reviewer #1: Yes

Reviewer #2: Yes

6. Review Comments to the Author

Reviewer #1: The authors appropriately address the questions and comments made by the reviewers. This has strengthened the manuscript.

Reviewer #2: (No Response)

7. PLOS authors have the option to publish the peer review history of their article (what does this mean?). If published, this will include your full peer review and any attached files.

Reviewer #1: No

Reviewer #2: No

---

## [Editor Report · Acceptance letter]

10 Feb 2022

PONE-D-21-14137R1 

The Association of Health Insurance and Race with Treatment and Survival in Patients with Metastatic Colorectal Cancer 

Dear Dr. Snyder:

I'm pleased to inform you that your manuscript has been deemed suitable for publication in PLOS ONE. Congratulations! Your manuscript is now with our production department. 

Kind regards, 

on behalf of

Dr. Girijesh Kumar Patel 

Academic Editor

PLOS ONE